# The Role of the Placental Enzyme Indoleamine 2,3-Dioxygenase in Normal and Abnormal Human Pregnancy

**DOI:** 10.3390/ijms25084577

**Published:** 2024-04-22

**Authors:** Yoshiki Kudo, Jun Sugimoto

**Affiliations:** Department of Obstetrics and Gynecology, Graduate School of Biomedical Sciences, Hiroshima University, Hiroshima 734-8551, Japan; juns@hiroshima-u.ac.jp

**Keywords:** indoleamine 2,3-dioxygenase, placenta, trophoblast, pre-eclampsia, placenta accreta spectrum, human pregnancy

## Abstract

The biologically significant phenomenon that the fetus can survive immune attacks from the mother has been demonstrated in mammals. The survival mechanism depends on the fetus and placenta actively defending themselves against attacks by maternal T cells, achieved through the localized depletion of the amino acid L-tryptophan by an enzyme called indoleamine 2,3-dioxygenase. These findings were entirely unexpected and pose important questions regarding diseases related to human pregnancy and their prevention during human pregnancy. Specifically, the role of this mechanism, as discovered in mice, in humans remains unknown, as does the extent to which impaired activation of this process contributes to major clinical diseases in humans. We have, thus, elucidated several key aspects of this enzyme expressed in the human placenta both in normal and abnormal human pregnancy. The questions addressed in this brief review are as follows: (1) localization and characteristics of human placental indoleamine 2,3-dioxygenas; (2) overall tryptophan catabolism in human pregnancy and a comparison of indoleamine 2,3-dioxygenase expression levels between normal and pre-eclamptic pregnancy; (3) controlling trophoblast invasion by indoleamine 2,3-dioxygenase and its relation to the pathogenesis of placenta accrete spectrum.

## 1. Introduction

The enzyme indoleamine 2,3-dioxygenase (IDO) (EC 1.13.11.42), widely expressed in a variety of tissues of mammals, catalyzes the oxidative cleavage of the essential amino acid L-tryptophan [1]. One tissue with particularly high activity is the human placenta [2]. With regard to the role of placental IDO, Munn et al. established the hypothesis in mouse that expression of this enzyme at the maternal–fetal interface regulates the maternal immune response to the fetal allograft and prevents its immunological rejection [3]. We showed that the same mechanism is available at the maternal–fetal interface of human pregnancy as in mice [4]. Thus, IDO-mediated localized depletion of tryptophan in human pregnancy can regulate proliferation of human peripheral blood mononuclear cells at the maternal–fetal interface [5].

The maternal syndrome of pre-eclampsia is a major complication of human pregnancy with significant morbidity and mortality. Although the cause of pre-eclampsia is still unknown, there is strong evidence linking the clinical condition to abnormalities in placental development, leading to increased oxidative stress [6,7] and dysfunction in maternal endothelial cells [8]. In addition to endothelial dysfunction, systemic activation of maternal inflammatory cell populations has been observed [9]. Munn et al.’s hypothesis suggests that placental IDO suppresses the maternal immune response against the fetus by depleting tryptophan at the maternal–fetal interface. It is possible to suggest that alterations in IDO activity or levels in the placenta may be involved in the pathogenesis of pre-eclampsia.

Placenta accreta spectrum (PAS) is a serious complication of pregnancy in which trophoblasts invade the myometrial layer. In normal implantation, trophoblast invasion into the maternal tissue is thought to be controlled by mechanisms in the decidual layer [10]. Immune cells, including macrophages and uterine natural killer cells, colonize the decidua and have been thought to be involved in the control of trophoblast invasion [11]. It has been suggested that tryptophan depletion by IDO expressed in decidual macrophages may be involved in inducing apoptosis of extravillous cytotrophoblast cells and controlling trophoblast invasion [12].

IDO is being investigated for its involvement not only in reproductive biology but also in various pathophysiological conditions of medical importance, such as inflammatory and autoimmune diseases, infectious diseases, neuropathology, cancer, and organ transplantation [13,14,15,16]. This brief review aims to focus on discussing the expression of IDO in the human placenta and its involvement in normal and abnormal human pregnancy.

## 2. Indoleamine 2,3-Dioxygenase (IDO)

The enzyme IDO is a heme-containing protein that catalyzes the oxidative cleavage of the pyrrole ring of the indole nucleus of various indoleamines derivatives (e.g., tryptophan, 5-hydroxytryptophan, tryptamine and serotonin) (Figure 1) upon the insertion of two oxygen atoms of molecular oxygen [1]. IDO is widely expressed in a variety of tissues of mammals such as rabbit [17], rat [18], mouse [19], and humans [2]. One tissue with particularly high activity is the human placenta [2]. Although the precise physiological roles of IDO are still unknown, the enzyme is induced under pathological conditions including virus infection [20], parasitic infestation [21], and tumor transplantation into allogenic animals [22,23], resulting in the rapid degradation of tryptophan to kynurenine in the infected or the tumor cells. Interferon-γ, which has potent immunomodulatory and antiproliferative effects, strongly induces the expression of the gene encoding IDO [24]. The antiproliferative effect of interferon-γ on tumor cells and its inhibitory effect on intracellular pathogens are thought to be, at least in part, due to the depletion of the essential amino acid L-tryptophan following induction of IDO.

A novel IDO isozyme has been identified and designated as indoleamine 2,3-dioxygenase-2 (IDO2) [25,26]. The original IDO is now referred to as indoleamine 2,3-dioxygenase-1 (IDO1). While these two proteins exhibit similar enzymatic activities, their distinct expression patterns within tissues suggest unique roles for each protein [25,27]. Due to the recent identification of IDO2, the physiological role of IDO has not been categorized into that of IDO1 and IDO2, and the term “IDO” in most published studies may imply a collective functional activity of IDO.

With regard to the role of placental IDO, Munn et al. [3] proposed the hypothesis that expression of this enzyme at the maternal–fetal interface is crucial to avoid immune rejection of the fetal allograft. To examine this hypothesis, they treated pregnant mice (carrying syngeneic or allogeneic fetuses) with 1-methyl-tryptophan (Figure 1), a pharmacological agent that inhibits IDO activity [28]. They observed that treating pregnant mice with the inhibitor resulted in rapid T cell-induced rejection of allogeneic concepti (but not syngeneic ones). However, fetal allograft rejection was not observed when RAG-1 (recombination activating gene 1)-deficient mothers, unable to develop lymphocytes, were treated with 1-methyl-tryptophan. They also demonstrated that adoptive transfer of splenocytes from genetically modified mice caused fetal allograft rejection in these mothers lacking lymphocytes. This rejection occurred when the cells specifically targeted a paternally inherited fetal MHC class I alloantigen. They next showed that massive deposition of complement and hemorrhagic necrosis occurred at the maternal–fetal interface when pregnant mice carrying an allogeneic fetus were exposed to 1-methyl-tryptophan and that this inflammation is driven by T cell recognition of fetal antigens [29]. Moreover, this complement deposition and fetal allograft rejection occurred in the absence of maternal B cells, suggesting that complement activation was antibody-independent. Thus, by catabolizing tryptophan, the mouse conceptus suppresses T cell activity and defends itself against rejection. However, several issues arise from their hypothesis, such as how the developing fetus is well supplied with tryptophan when cells at the maternal–fetal interface degrade it. It is possible that IDO-mediated tryptophan catabolism may produce an immunosuppressive metabolite. Quinolinic acid, for example, is a potent neuroexcitatory toxin that putatively could serve as a mediator of cell destruction in a variety of neurodegenerative disorders [30]. Additionally, IDO-mediated tryptophan catabolism consumes oxygen radicals [1] and this might influence T cell responsiveness.

In human tissue, IDO is detectable immunohistochemically from day 6 human blastocysts and thereafter throughout pregnancy in syncytiotrophoblasts, cytotrophoblasts, and macrophages in the villous stroma and in the fetal membranes [4]. Other tryptophan catabolizing enzymes in humans are tryptophan 2,3-dioxygenase (EC 1.13.11.11) and tryptophan hydroxylase (EC 1.14.16.4) (Figure 2). Tryptophan 2,3-dioxygenase is found only in the liver and is induced by administration of tryptophan. Tryptophan hydroxylase, which is expressed by neuronal cells of the brain and adrenal gland, is the first step in the synthesis pathway for serotonin and its derivatives.

## 3. Paradox of Immunological Tolerance toward the Fetus during Pregnancy

It is recognized as paradoxical that a genetically different mammalian conceptus, expressing both paternal and maternal gene products, evades any maternal immune response during pregnancy. Medawar [31] proposed three hypotheses to explain this paradox of maternal immunological tolerance to the fetus: anatomical separation of mother and fetus, antigenic immaturity of the fetus, and suppression or modification of the maternal immune system. The first two hypotheses cannot fully explain fetal allograft survival, as the maternal–fetal interface is not an absolute barrier and fetal cells can migrate into the maternal circulation [32]. The second hypothesis is also nearly ruled out because both fetal and placental cells express major histocompatibility complex molecules (MHCs) [33,34]. Therefore, major interest has been focused on the third mechanism, where the survival of the genetically different fetus depends on active defense by the fetus and placenta against attack by maternal immune cells. The following are among the processes proposed which allow the maternal immune system to be tolerant of the fetal allograft. Progesterone, which is synthesized at high rates in the placenta, has been shown to reduce the immune response [35]. Immunosuppressive molecules may be expressed on the placental surface, e.g., the nonclassical MHC class I antigen human leukocyte antigen (HLA)-G is expressed by the syncytiotrophoblast. Binding of natural killer cells to this molecule may downregulate its activity [36], e.g., the Fas ligand (or CD95 ligand) expressed on the surface of the syncytiotrophoblast induces the apoptosis of activated T cell by binding to its Fas receptor (or CD95) [37,38]. It has also been suggested that the population of helper T cells, namely, Th1 and Th2 balance, has an implication for maintaining normal pregnancy which is chiefly viewed as Th2-type T helper cell response predominant [39,40]. Munn et al. [3] added IDO as a new candidate to the list of potential immunosuppressive mechanisms in pregnancy. These authors’ hypothesis is that in the mouse, the expression of IDO, a major tryptophan-catabolizing enzyme, in the placenta is crucial to prevent immunological rejection of the fetal allograft. They suggested that T cells are inhibited by a mechanism involving IDO-dependent localized depletion of tryptophan at the site of placentation.

However, it seems likely that by themselves, none of these mechanisms will be sufficient to explain the maternal immune tolerance to the allogeneic fetus. Integration of these and other mechanisms is almost certainly required for the success of mammalian viviparity.

## 4. Localization and Characterization of Human Placental IDO

We have previously demonstrated that in the first trimester, placenta immunohistochemical staining for IDO was found in syncytiotrophoblast, extravillous cytotrophoblast and macrophages in the villous stroma [4]. Staining was also seen in the glandular epithelium and stromal cells of the first trimester decidua [41]. Sedlmayer et al. [42] also found IDO to be strongly expressed in the glandular epithelium with some positive cells in the decidual stroma. They did, however, find that staining of the syncytiotrophoblast was comparatively rare. Interestingly, both of these studies used the same monoclonal antibody as IDO [43]. The question of trophoblastic expression of IDO has been a continuous controversial issue.

As mentioned above, a novel IDO isozyme was identified and assigned the name IDO2, while the original IDO is referred to as IDO1 [25,26]. We therefore mapped the immunohistochemical distribution of the IDO1 and IDO2 proteins at the human maternal–fetal interface using a rare early pregnancy sample from a women at seven weeks of gestation who underwent hysterectomy for cervical cancer with gestational sac in utero [44].

Obvious expression of IDO1 and IDO2 is seen at the maternal–fetal interface in the seven-week placenta (Figure 3). There is strong expression of IDO1 on the glandular epithelium, endothelium of spiral artery, and on CD68-positive macrophages in the decidua with little immunopositivity in the villous core (fetal vessel endothelial cells). The cellular expression of IDO2 shows obvious syncytiotrophoblastic expression. The results of the immunohistochemistry mentioned above were obtained using the antibody to either human IDO1 or IDO2 raised against a recombinant protein for each enzyme, respectively. The previous studies, including ours, were conducted before discovery of IDO2, and the authors used the monoclonal antibody to IDO prepared by Takikawa et al. [43] for immunohistochemical analysis [4,42,45,46]. Since Takikawa’s antibody to human IDO was raised by using human IDO protein purified from human placenta, it is possible that this antibody may react with both IDO1 and IDO2 proteins, potentially accounting for the discrepant result described above.

Expression of IDO2 in syncytiotrophoblasts is also confirmed functionally by immunohistochemistry of kynurenine. Kynurenine is the immediate downstream product of IDO-mediated tryptophan catabolism. Immunostaining for kynurenine is found in the syncytiotrophoblast and in the glandular epithelium and some cells (extravillous cytotrophoblasts) in the decidua. This kynurenine immunoreactivity has essentially the same localization as that of IDO1 and IDO2, indicating that the IDO proteins appear to be functional enzymes. Interferon-γ added to placental villous explant culture markedly stimulated expression level of both mRNA and immunoreactivity of IDO1. In contrast, IFN-γ showed no stimulatory effect on both mRNA expression level and immunoreactivity of IDO2 [44].

## 5. Tryptophan Catabolism by Placental IDO in Human Pregnancy: A Comparison of Normal Pregnancy and Pre-Eclampsia

In healthy pregnant women tryptophan concentration in blood declines progressively as a function of gestational age, and the decrease has been postulated to be related to the maternal immune response to the fetus [47,48]. Considering that the maternal syndrome of pre-eclampsia is distinguished by an excessive systemic inflammatory response induced by pregnancy [49], it is plausible that placental IDO may exhibit reduced efficacy in modulating local and, consequently, circulating tryptophan concentrations in these pregnancies. We investigated placental IDO activity and indices of tryptophan catabolism, specifically the ratios of the IDO product (plasma kynurenine) to substrate (plasma tryptophan), in pregnant women with and without pre-eclampsia, as well as in nonpregnant women of reproductive age [50].

Table 1 summarizes the results of high-performance liquid chromatography analysis of tryptophan and kynurenine concentrations in plasma. Samples from 12 women with pre-eclampsia, 12 appropriately matched women with normal pregnancy, and 9 healthy nonpregnant women were studied. Pre-eclampsia was defined as new hypertension (diastolic > 90 mmHg) accompanied by new proteinuria after 20 weeks of gestation. The normal pregnant and pre-eclamptic women were well matched for age, parity, and gestation age. The concentrations of plasma tryptophan were significantly lower in pregnant women compared to nonpregnant women, regardless of their pregnancy status. Furthermore, women with normal pregnancies exhibited significantly lower tryptophan concentrations than those with pre-eclampsia. Plasma kynurenine concentrations showed the converse patterns; hence, the ratios of plasma kynurenine to tryptophan, an index of tryptophan catabolism, were significantly increased in normal pregnant women compared either to women who were not pregnant or to women with pre-eclampsia. The ratios of kynurenine to tryptophan for women with pre-eclampsia were not different from those for women who were not pregnant.

The IDO activities in fresh placental villous tissue were significantly lower in pre-eclampsia compared to normal pregnancy (normal pregnancy, 0.48 ± 0.06 nmol/mg protein/min; pre-eclampsia, 0.29 ± 0.04 nmol/mg protein/min). When villous tissue explants were cultured for 36 h with interferon-γ at 1000 unit mL^−1^ (the condition maximally stimulating IDO activity), both IDO activity (normal pregnancy, 1.58 ± 0.04; pre-eclampsia, 0.76 ± 0.14 nmol/mg protein/min) and the percentage stimulation (normal pregnancy, 329.1 ± 43.3; pre-eclampsia, 258.8 ± 48.3) were still significantly lower in villous tissue from pre-eclampsia than was found for tissue from normal pregnancy. Consequently tryptophan concentration in the conditioned medium was higher when culture had been conducted using villous tissue from pre-eclampsia as compared with that from normal pregnancy either in the presence or in the absence of interferon-γ. When peripheral blood mononuclear cells were cultured in the conditioned medium of villous tissue explants, inhibition of peripheral blood mononuclear cell proliferation activity was less with medium previously conditioned by culture of villous explants from pre-eclampsia [50]. These differences are more marked in media conditioned in the presence of interferon-γ. Flow cytometric analysis showed that proliferation of CD4 positive T helper lymphocytes is specifically suppressed by IDO-mediated tryptophan depletion [5].

The level of IDO mRNA expression was decreased by 44.2% in fresh villous tissue from pre-eclampsia compared with that from normal pregnancy [50]. The regulation of the IDO mRNA expression by interferon-γ in the pre-eclamptic placenta is also disturbed. Specifically, interferon-γ does not induce the expression of the gene in the same way as it does in normal pregnancies (normal pregnancy, 2.1-fold; pre-eclampsia, 1.7-fold), although in pre-eclampsia, interferon-γ does activate expression of other genes (i.e., signal transducer and activator of transcription1 and tryptophanyl-tRNA synthetase) normally. Signal transducer and activator of transcription1 is required for interferon-γ-dependent transcription [51], and tryptophanyl-tRNA synthetase expression is also induced by interferon-γ through the same pathway as IDO [52]. These observations specifically implicate the IDO gene in the etiology of pre-eclampsia. The results of sequencing of the IDO gene in babies from pre-eclamptic pregnancies show that sequences corresponding to the promoter region in which signal transducer and activator of transcription1 binding site, interferon-γ activation site [53], or interferon-stimulated response element [54] is present, and exon 1 to exon 10 of the human IDO gene [55] appear normal.

## 6. Placental IDO and Trophoblast Invasion: Implications for the Pathogenesis of PAS

In normal implantation, trophoblast invasion into the maternal tissue is thought to be controlled by mechanisms in the decidual layer [10]. During implantation, cytotrophoblast cells of chorionic villi contact the maternal decidua and differentiate into extravillous cytotrophoblast cells and invade across the maternal decidua as interstitial extravillous cytotrophoblasts. Factors responsible for regulating the extent of extravillous cytotrophoblast invasion are poorly understood [11]. In vitro experiments demonstrated that IDO expressed by macrophages have the potential to actively induce apoptosis in extravillous cytotrophoblast cells by IDO-mediated tryptophan depletion [12]. They suggested that tryptophan depletion by IDO expressed in decidual macrophages in vivo also may be involved in inducing apoptosis and controlling trophoblast invasion.

It is well recognized that a cesarean delivery is one of the predisposing factors for placental pathologies, including PAS, in subsequent pregnancies. Placental implantation at the site of a previous cesarean scar is an extremely serious complication of pregnancy in which trophoblasts invade the myometrial layer, resulting in placenta accreta and percreta. We therefore redefine in vivo IDO localization using a rare early pregnancy sample from a woman at 9 weeks of gestation who underwent hysterectomy for placental implantation on the scar of a previous cesarean section [56]. This case, thus, sheds light on the pathophysiology of placental implantation over a previous cesarean scar.

The section of this sample includes the placental–decidual interface as well as myometrial tissue, and at a scar of cesarean section, decidual tissue was disrupted between placental villous tissue and myometrium (Figure 4A). These were used for immunohistochemical analysis. IDO immunoreactivity was seen to be strongly expressed on the glandular epithelium in the decidua (Figure 4B(a)). Some IDO-positive cells were also seen in the decidual stroma, which are CD68-positive macrophages. HLA-G-positive extravillous cytotrophoblast cells were observed only within the decidua and they did not invade the myometrium; however, extravillous cytotrophoblast cells obviously invaded the myometrium at a site of cesarean scar where decidual tissue was not present (Figure 4B(b)).

To further confirm the possibility that IDO regulates trophoblast invasion, we conducted a co-culture experiment with trophoblast cells (HTR-8/SVneo cells) and cells (Ishikawa cells) that overexpressed IDO. This experiment utilized a transwell migration assay, involving two medium-filled chambers separated by a porous membrane. When trophoblast HTR-8 cells were co-cultured with cells overexpressing IDO, trophoblast migration was suppressed compared to when trophoblast cells were co-cultured with cells not expressing IDO. This co-culture experiment also suggests that IDO expressed in the decidua may be involved in controlling trophoblast invasion.

## 7. Conclusions

Experiments described in this short review were conducted to investigate the role of the placental enzyme IDO in normal and abnormal human pregnancy. We demonstrated the obvious expression of IDO1 and IDO2, a more recently identified novel isoform of IDO, at the human maternal–fetal interface at seven weeks of gestation obtained from hysterectomy for cervical cancer with a pregnancy in situ. The findings of our study, that the distribution of IDO2 are dissimilar to those of IDO1 at the human maternal–fetal interface, suggest involvement of IDO2 in normal human pregnancy. We would like to suggest that IDO2 expressed in the syncytiotrophoblasts is responsible for regulating maternal immune response to the fetal allograft at the maternal–fetal interface. That the IDO2 may be involved in immune evasion by tumor has also been suggested in tumor biology [26].

Our study showed that in pre-eclamptic placenta there is suppressed expression of placental IDO and its regulation by interferon-γ are disturbed. Consequently IDO-mediated inhibition of peripheral blood mononuclear cell proliferation activity is less in pre-eclampsia. In the peripheral blood of women with pre-eclampsia leukocyte (including lymphocyte) activation is known to be exaggerated compared with that in normal pregnancy [49,57]. These findings therefore provide evidence for a connection between abnormal regulation of the maternal inflammatory response in pre-eclampsia and disrupted IDO-mediated manipulation of tryptophan at the maternal-fetal interface. Flow cytometry analysis showed that it is CD4-positive lymphocytes which are specifically influenced by IDO-mediated tryptophan depletion [5]. Helper T cells are major source of cytokines and further subdivided into Th1 and Th2. Healthy pregnancy is chiefly regarded as Th2 dominant phenomena; a strong Th2 response is necessary to modify the Th1 cellular response in utero to reduce the risk of miscarriage [58]; Th1 is predominant in pre-eclamptic pregnancy [59]. It is possible to speculate that disturbed regulation of IDO by cytokines is related to abnormal setting of the pathway of T helper cell differentiation which is likely to underlie the abnormal inflammatory response in pre-eclampsia.

The reason for the reduced expression level of IDO in pre-eclamptic placenta has yet to be determined. However, it is probable that the elevated plasma tryptophan concentration in women with pre-eclampsia, along with tryptophan-dependent proliferation of lymphocytes, suggests a causative link between suppressed IDO expression and the exaggerated inflammatory response observed in this maternal syndrome. A proposal for the role of placental IDO in suppressing the maternal systemic inflammatory response during human pregnancy and in the pathogenesis of pre-eclampsia is illustrated in Figure 5 [50]. If the IDO-mediated mechanism were not present, the systemic inflammatory response would be exaggerated. In normal pregnancy, placental IDO suppresses this exaggerated inflammatory state to a normal level. However, in pre-eclampsia, decreased placental IDO activity is not sufficient to suppress the immune inflammatory response to the level found in normal pregnancy. These assumptions also raise the possibility that novel therapeutic interventions in pre-eclampsia might be effectively focused on manipulating plasma tryptophan concentration.

It has been suggested in an in vitro model that IDO-mediated tryptophan depletion induces apoptosis of extravillous trophoblast cells, thereby controlling trophoblast invasion and leading to normal placentation [12]. We have shown, using a cesarean scar pregnancy specimen, that IDO expressed in the decidua may control extravillous cytotrophoblast invasion at the site of implantation, and absence of its expression may be involved in the pathogenesis of over-invaded placenta [56]. We identified certain features of cesarean scar pregnancy that could help in understanding the mechanism of PAS. However, it is uncertain if cesarean scar pregnancy is representative of all PAS cases, as our data are based on a single case. Further research is required, but this case allowed us to add IDO to the list of mechanisms causing abnormal placental implantation.

Although there are still fundamental questions about the role of IDO in human pregnancy, we think that the data described in this short review by studying IDO expression at the maternal–fetal interface and its involvement in the pathogenesis of pre-eclampsia and PAS may help address some of these. The most straightforward experimental strategy to delineate the significance of IDO in mammalian reproductive physiology and the pathogenesis of pathological pregnancy might be the use of knockout animals for this enzyme. In humans, no nucleotide polymorphism of the gene encoding these enzymes has been reported; this needs to be studied, particularly in relation to pathological human pregnancy.

## Figures and Tables

**Figure 1 ijms-25-04577-f001:**
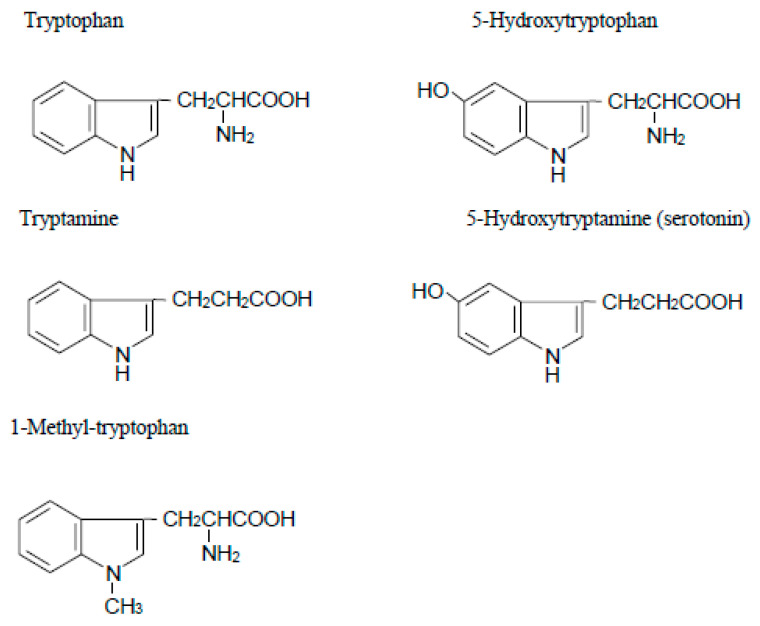
Structure of tryptophan analogues.

**Figure 2 ijms-25-04577-f002:**
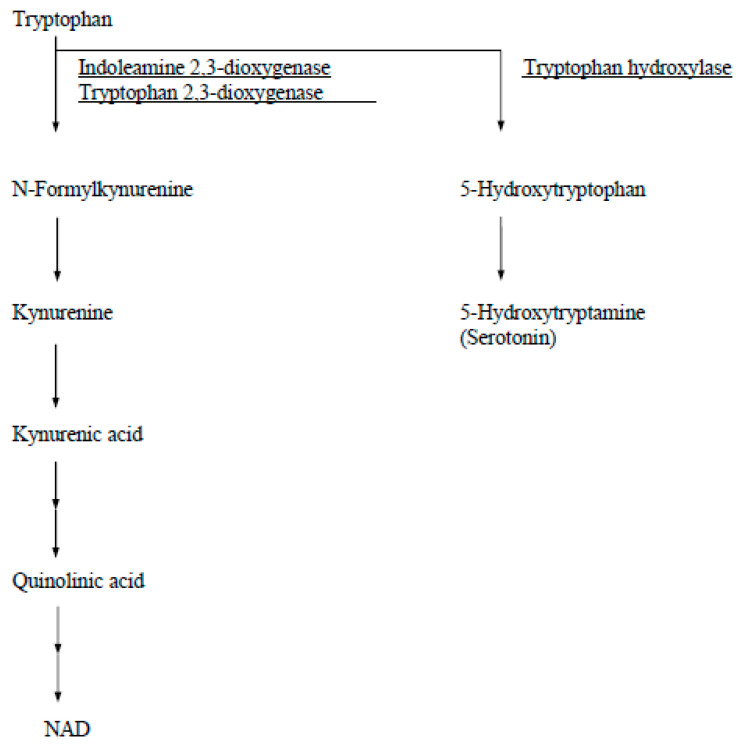
A diagram of catabolic pathways for tryptophan.

**Figure 3 ijms-25-04577-f003:**
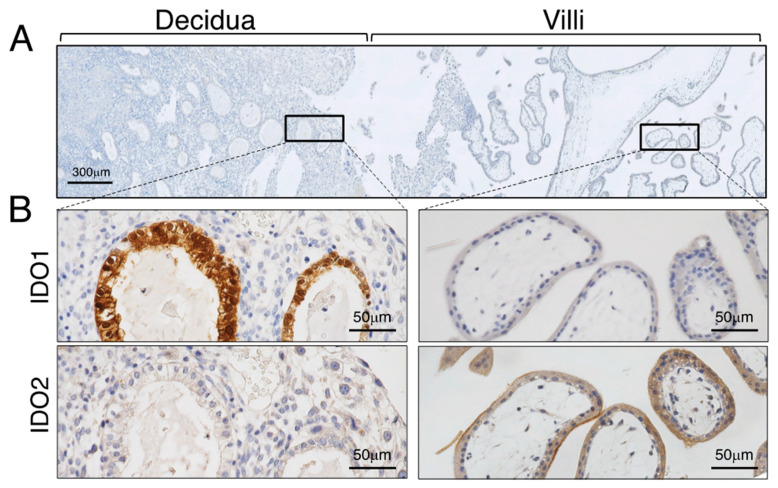
Immunohistochemistry of IDO at the maternal–fetal interface of seven weeks of gestation. (**A**) Implantation site of seven weeks of gestation. (**B**) Immunohistochemical localization of IDO. Reproduced from Kudo et al. [44]. Scales for images are as indicated.

**Figure 4 ijms-25-04577-f004:**
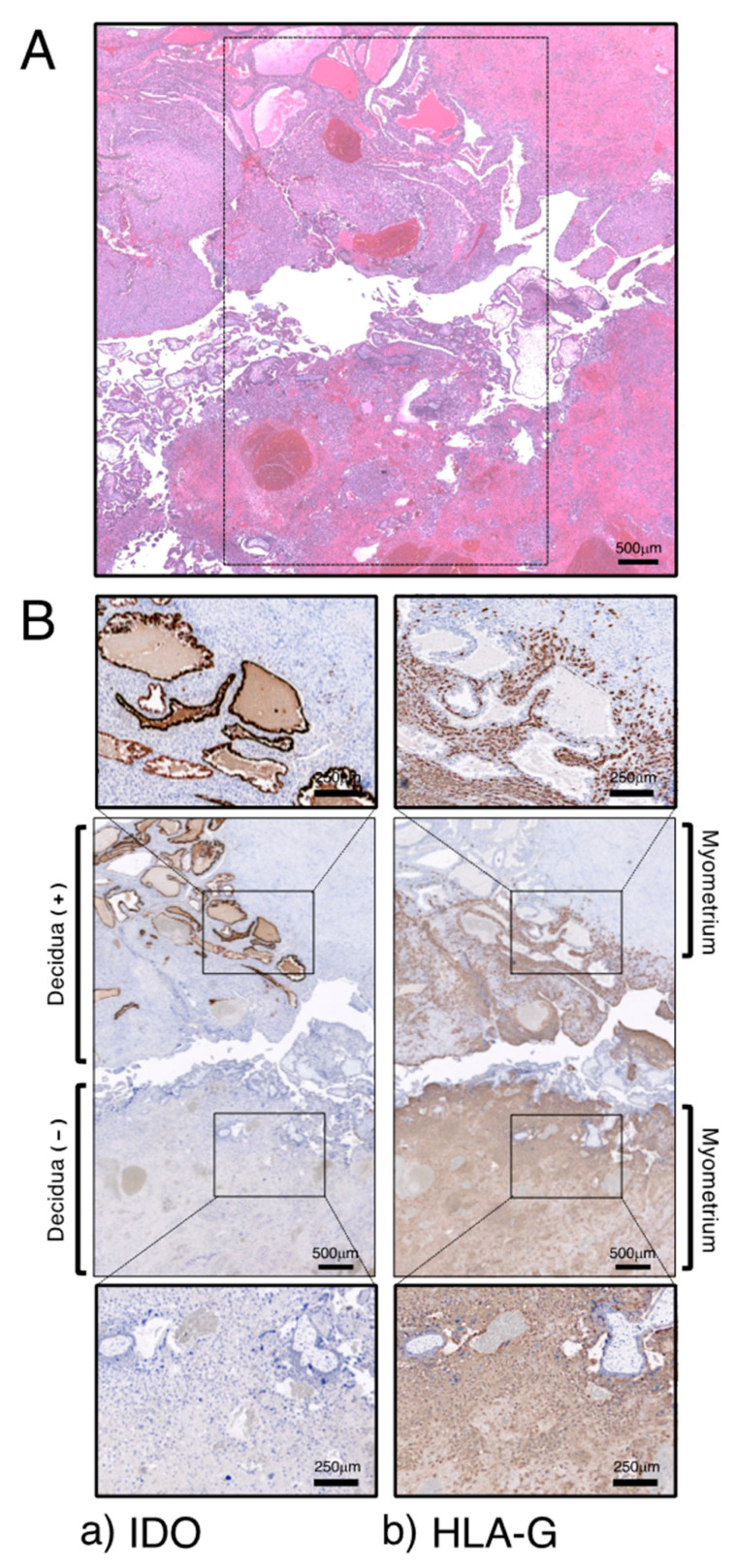
Immunohistochemical localization of IDO in decidua. (**A**) HE staining. (**B**) Immunostaining for IDO (**a**) and HLA-G (**b**). Reproduced from Kudo et al. [56]. Scales for images are as indicated.

**Figure 5 ijms-25-04577-f005:**
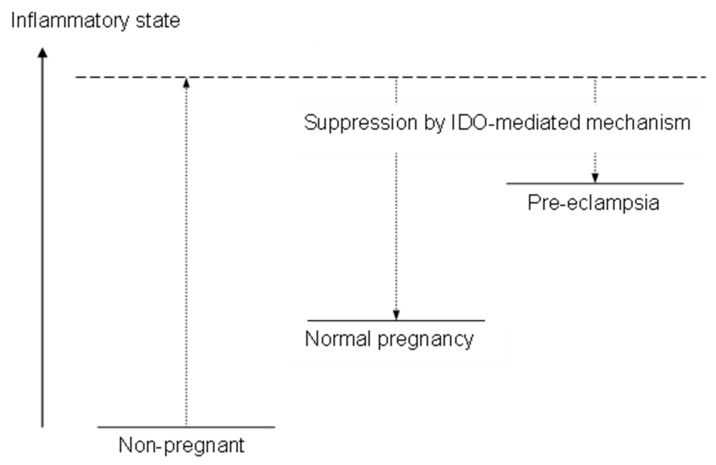
A schematic representation of the role of placental IDO in human pregnancy. Reproduced from Kudo et al. [50].

**Table 1 ijms-25-04577-t001:** Tryptophan and kynurenine concentrations and the ratio of kynurenine to tryptophan [50].

	Normal Pregnancy (n = 12)	Pre-Eclampsia (n = 12)	Nonpregnant (n = 9)
Tryptophan (µM)	32.7 ± 4.8	42.8 ± 6.9 *	53.0 ± 9.8 ^†,‡^
Kynurenine (µM)	1.12 ± 0.17	1.02 ± 0.22	1.17 ± 0.28
Kynurenine/tryptophan	0.034 ± 0.004	0.024 ± 0.005 *	0.022 ± 0.004 ^‡^

* *p* < 0.001 compared with normal pregnancy (Wilcoxon test). ^†^ *p* < 0.001 compared with pre-eclampsia (Mann–Whitney *U*-test). ^‡^ *p* < 0.001 compared with normal pregnancy (Mann–Whitney *U*-test). Values are given as mean ± SD.

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
