# Peer review of "The Role of the Placental Enzyme Indoleamine 2,3-Dioxygenase in Normal and Abnormal Human Pregnancy"

_ijms, 2024, doi:10.3390/ijms25084577_

Round 1
Reviewer 1 Report
Comments and Suggestions for Authors
This review explores the intricate role of the placental enzyme indoleamine 2,3-dioxygenase (IDO) in human pregnancy, focusing on its involvement in maternal-fetal immune tolerance and pregnancy-related complications such as pre-eclampsia and placenta accreta spectrum disorders. By elucidating IDO's function in tryptophan metabolism and its impact on immune regulation at the maternal-fetal interface, the review sheds light on potential therapeutic targets and diagnostic markers for pregnancy disorders. Through a comprehensive analysis of IDO's molecular mechanisms and clinical implications, the review provides valuable insights into the complex interplay between maternal immune responses and placental physiology during pregnancy.
Some improvements can be made to improve the clarity and readability of the text.
Comments:
- This review could benefit from a more concise and structured presentation. It seems like the authors compacted lots of information in this article without well-organization. For example, “Chapter 2.1. IDO” should include: a) IDO working pathway (tryptophan metabolism), b) IDO’s role in immune regulation, c) IDO’s role in allergy and inflammation, d) IDO inhibitor, e) IDO as a potential biomarker. This suggestion I mentioned is just an example and it is more than welcome if you have other ideas to organize the whole paper better. The whole paper’s structure needs to be reorganized.
- Another suggestion is to provide a clearer roadmap for readers to get a whole picture of this paper. (A summarized graph for this review would be better, Figure 3 is relatively rough or simple).
- Providing more context in discussion about the limitations and future directions would be valuable.
Moderate editing of English language required
Author Response
Thank you for your helpful report which we found very useful.
1) As suggested, we have added the structure of tryptophan analogues and IDO inhibitor (1-methyl-tryptophan) (Figure 1), and the tryptophan metabolic pathway (Figure 2) to Chapter 2.1. IDO. Additionally, we have described the involvement of IDO in human diseases other than reproductive biology in the Introduction section as per your suggestion (page 2, line 70 - 75).
2) We have added a discussion on the limitations (page 11, line 402 - 407 and page 11 - 12, line 422 - 426) and future prospects (page 12, line 430 - 432) in the Discussion section.
Reviewer 2 Report
Comments and Suggestions for Authors
Dear Authors,
You may find some comments in the attached PDF. Please reformulate the abstract and review the entire manuscript and try to decide if this paper is an original research since you are emphasizing specifically your own research or is a review.

Comments on the Quality of English LanguageTry to reformulate the phrases that are to long and very difficult to read. I am sure that by doing that, your manuscript will be more readable and more accessible.
Author Response
We are grateful to the referee for your positive comments.
1) We have accepted and incorporated all the comments provided in the PDF, and we have rewritten accordingly. We also have modified the Abstract as suggested.
2) We have considered the type of manuscript, but we would like to keep it as a review article.
3) As suggested, we have attempted to rephrase the lengthy sentences to make it easier to read. Thank you!